# Current Knowledge on Snake Dry Bites

**DOI:** 10.3390/toxins12110668

**Published:** 2020-10-22

**Authors:** Manuela B. Pucca, Cecilie Knudsen, Isadora S. Oliveira, Charlotte Rimbault, Felipe A. Cerni, Fan Hui Wen, Jacqueline Sachett, Marco A. Sartim, Andreas H. Laustsen, Wuelton M. Monteiro

**Affiliations:** 1Medical School, Federal University of Roraima, Boa Vista 69310-000, Roraima, Brazil; manupucca@hotmail.com; 2Department of Biotechnology and Biomedicine, Technical University of Denmark, DK-2800 Kongens Lyngby, Denmark; cecknu@dtu.dk (C.K.); charim@dtu.dk (C.R.); ahola@bio.dtu.dk (A.H.L.); 3Bioporto Diagnostics A/S, DK-2900 Hellerup, Denmark; 4Department of BioMolecular Sciences, School of Pharmaceutical Sciences of Ribeirão Preto, University of São Paulo, Ribeirão Preto, São Paulo 14049-900, Brazil; isadora_so@yahoo.com (I.S.O.); felipe_cerni@hotmail.com (F.A.C.); 5Butantan Institute, São Paulo 05503-900, Brazil; fan.hui@butantan.gov.br; 6Department of Medicine and Nursing, School of Health Sciences, Amazonas State University, Manaus 69065-001, Amazonas, Brazil; jac.sachett@gmail.com; 7Department of Teaching and Research, Alfredo da Matta Foundation, Manaus 69065-130, Amazonas, Brazil; 8Department of Teaching and Research, Dr. Heitor Vieira Dourado Tropical Medicine Foundation, Manaus 69040-000, Amazonas, Brazil; marcosartim@hotmail.com; 9Institute of Biological Sciences, Amazonas Federal University, Manaus 69067-005, Amazonas, Brazil

**Keywords:** dry bites, venom, non-envenoming, snakebite, antivenom, asymptomatic envenoming

## Abstract

Snake ‘dry bites’ are characterized by the absence of venom being injected into the victim during a snakebite incident. The dry bite mechanism and diagnosis are quite complex, and the lack of envenoming symptoms in these cases may be misinterpreted as a miraculous treatment or as proof that the bite from the perpetrating snake species is rather harmless. The circumstances of dry bites and their clinical diagnosis are not well-explored in the literature, which may lead to ambiguity amongst treating personnel about whether antivenom is indicated or not. Here, the epidemiology and recorded history of dry bites are reviewed, and the clinical knowledge on the dry bite phenomenon is presented and discussed. Finally, this review proposes a diagnostic and therapeutic protocol to assist medical care after snake dry bites, aiming to improve patient outcomes.

## 1. Introduction

Every year, about 5.4 million snakebites occur worldwide. These cause up to 2.7 million envenomings, almost 138,000 deaths, and 400,000 cases of sequelae or disability [1,2]. This serious public health problem is a neglected, occupational disease in subtropical and tropical countries in Asia, Africa, and Latin America [1]. Every year, two million snakebites occur in Asia, with India presenting more than 46,000 deaths each year [1,3]. In Africa, snakebites are estimated to cause 435,000 to 580,000 accidents annually, with a range of 7000 to 32,000 deaths in sub-Saharan Africa, of which 3500 to 5400 deaths occur in West Africa [1,4,5,6,7]. In Latin America, there are 137,000 to 150,000 snakebite reports and 3400 to 5000 deaths per year [8]. In developed countries in North America and Europe, these numbers are much lower. In North America, 3800 to 6500 annual cases with up to 15 deaths are reported, while Europe experiences around 7500 cases and up to five deaths per year [4,7,8,9,10].

Snakebites are associated with poverty, and the most at-risk groups include farmers and their families, fishermen, hunters, woodcutters, indigenous people, and indigents, as well as people who do not have access to adequate health and educational systems [6,8,11]. Children and pregnant women are also considered risk groups for this neglected disease, with children often being bitten when they are playing outside, as curiosity might entice them to touch snakes. Due to the immaturity of their immune systems and low body weights compared to adults, envenomings in children are often severe. In pregnant women, snakebite envenoming has been documented to occasionally cause abortions, mainly due to hemorrhage [8,12].

In May 2019, the World Health Organization (WHO) launched a program to prevent and control snakebite incidents via (amongst other strategies) improved access to effective and safe treatment for the most affected communities, thereby aiming to reduce snakebite mortality and morbidity by 50% by 2030. In addition, the WHO encourages the search for new treatments, diagnostics, and preventative measures that can lead to faster recovery of snakebite victims or avoidance of victims being bitten in the first place [13].

No venomous snake is large enough to consider human beings as prey, and the main reason for human snakebite envenomings is that the snake defends itself against what it considers an imposing threat. However, as venom is metabolically costly to produce [14] and may not in itself immediately deter an imposing human or predator, as most venoms need time to exert their toxic effects, snakes may benefit from delivering warning bites devoid of venom (‘dry bites’) to predators and threats, thereby saving their venom for future prey. Thus, in many encounters, where snakes bite human victims, the victim may fortuitously only receive a dry bite. For example, a study conducted over three years at the Toxinology and Toxicology Unit of the General Hospital of the Central Province of Sri Lanka in Peradeniya demonstrated that in over 776 snakebite admissions, 86% of the patients had received a bite in which no venom had been injected [15]. It can, however, be difficult to determine whether a bite is ‘dry’ or ‘wet’, as a bite from any animal will often cause inflammation and swelling. This may complicate clinical diagnosis, making it more challenging to decide early on whether or not a snakebite victim needs antivenom. In this article, we present the anatomy of the snake venom apparatus, review the current knowledge on snake dry bites, and discuss the diagnostic and clinical implications of these bites.

## 2. Snake Venom Apparatus and Venom Production 

The anatomy of venomous snakes is diverse, but some aspects are universal [16]. All venomous snakes have similar venom delivery apparatuses comprised of a set of fangs, venom ducts, a pair of accessory glands, and a pair of postorbital glands, in which venom is produced. These venom glands are comprised of three major cell types: basal cells, conical mitochondria-rich cells, and secretory cells, which produce the venom (Figure 1). Venom is carried from the gland to the fang by a duct that flows through an accessory gland (absent in some snake families). It has been postulated that this accessory gland may be a site of activation of venom components [17,18,19], however, protein components that have been added to the venom after passing through the accessory gland, compared with venom extracted from the main venom glands, are yet to be identified. Finally, the apparatus ends with fangs, which are cone-shaped, tapered, and are usually curved, essentially making them into hollow venom delivering tubes [20]. Fangs can be replaced if lost or damaged, and some species even have reserve fangs that remain in a suspended and immature state until they are stimulated to develop [21]. The fangs are specialized for venom delivery, and some fangs have specialized even further to allow the snake to spit venom. The fangs can occupy various positions on the upper jaw but are always found on the maxilla. Additionally, in venomous colubrids (*sensu lato*) and elapids, the venom fang is attached to a stable maxillary bone, and for this reason it is always erect. In viperid snakes, the maxilla is capable of rotating, enabling the fang to be erected during a bite or laid parallel to the jaw when in the relaxed state [21]. Advanced snakes, which utilize venom for prey capture (colubrids, Viperidae, Elapidae, and Actraspidae) [20], are often referred to by the position of their fangs as either rear- or front-fanged snakes. The vast majority of snakes that are medically important to humans are front-fanged (Viperidae and Elapidae) [17,20].

The initiation of venom production appears to be controlled by the sympathetic nervous system in the venom glands of rattlesnakes (Viperidae), which have been taken as an excellent model system for the study of the synthesis, secretion, and storage of toxic proteins [22,23]. Venom production in both adult and juvenile specimens occurs very rapidly. The protein re-synthesis triggered by venom expulsion peaks between days 3–7 of the cycle of venom replenishment [24]. Protein synthesis is thought to be maintained at a high rate until completion, and studies have suggested that secretory cells can remain active for up to 30 to 60 days post venom extraction, indicating that the complete cycle of venom synthesis may be longer than previously expected [25]. After synthesis, the venom is stored in the basal lumen and ductules of the venom gland and is therefore available when needed [17].

Venom injection is a form of venom expulsion, in which the venom is discharged while the fang is imbedded in the tissue of the prey. It is characterized by variation in both the volume and pressure of the venom. In Viperidae, Elapidae, and Atractaspidinae snakes, the venom glands are enclosed in a fibrous sheath that facilitates the attachment of muscles. This muscularization of the venom glands allows the ejection of venom from the glands into prey in a high-pressure manner by contraction of the compressor muscle [26,27,28]. In viperids, the ability to rotate the maxilla offers an additional layer of control, as venom flowing through the distal portion of the venom ducts is affected by the degree of fang erection, at least until a threshold of approximately 60° degrees is reached. This enables the fang to be erected in the event of an attack or laid parallel to the jaw when in the relaxed state, as opposed to venomous elapids, in which the venom fang is attached to a stable maxillary bone and for this reason is always erect [26]. As such, viperids possess the most morphologically specialized and efficient venom delivery systems of all venomous snakes.

## 3. History of Snake Dry Bites

In London in 1892, a 30-year-old male was bitten by his pet South American rattlesnake (*Crotalus durissus*) [29]. The patient sought medical assistance and was examined by a physician one hour after the bite. The medical examination confirmed the presence of one fang puncture mark, and the physician applied a tourniquet, nitric acid, and potassium permanganate to the bite site (these were contemporary methods used for chemical cauterization and venom inhibition, respectively [30,31]). The patient neither presented evident signs and symptoms of envenoming while staying overnight at the hospital nor in the follow-up after discharge. In the physician words, “*I am inclined, however, to believe that either no poison had been injected at all, or so little as to cause no ill effects*” [29]. This case report possibly constitutes the first formal clinical report of a venomous snakebite accident without clinical manifestations. Since then, asymptomatic cases have been reported in epidemiological studies involving snakebites worldwide, combining reports from both venomous and non-venomous snakes. Bites that do not cause clinical manifestations were classified in the late 1950s and early 1960s as asymptomatic or grade 0, and they comprise bites with the presence of fang marks but minimal or no local or systemic signs and symptoms of envenoming [32,33]. These cases account for about 50% of bites globally, and for up to 80% of bites for some snake species [34].

The term ‘dry bites’ did not appear until the early 1980s, when it was defined as a venomous snakebite with no or negligible venom injection, characterized by a lack of clinical manifestations of envenoming [35]. The amount of snake venom injected at the site of the bite is considered the key factor influencing the severity of a venomous snakebite. The propensity of snakes to deliver dry bites has a multifactorial explanation, part of which is related to snake behavior. This was demonstrated by Kardong et al., who studied the biting habits of Northern Pacific rattlesnakes (*Crotalus viridis oreganus*) and found that often less venom was injected in the first strike of a series, and that defensive strikes resulted in more cases of dry bites compared to offensive (predatory) strikes [36]. Behavioral disparity is also reflected in the rates of dry bite cases reported for different venomous snake species [34,37]. Therefore, the dry bite definition represents an important hallmark in clinical snakebite investigations, considering its major medical importance and misestimated frequency rates compared to non-venomous snakebites. An example of this was reported by Walter et al. (2010) [38], who performed a retrospective epidemiological study concerning coral snakebites in the USA using data from 1983 to 2007. In that study, authors found a significant decrease in annual incidence rates of patients with no clinical manifestations of envenoming. These results can be explained by a lack of standardization of the definition of dry bites and their distinction from envenomed cases, especially mild envenomings, at the beginning of the period, and an improvement in surveillance over time [38].

It is important to diagnose dry bites in clinical settings as this diagnosis informs an adequate medical response, e.g., on whether antivenom is indicated or not. However, another important reason for dry bites to be diagnosed and reported concerns the spread of misinformation pertaining to so-called miraculous treatments. Traditional medicines and outdated treatment methods are still employed to combat snakebites in many parts of the world, whether alone or in conjunction with regular antivenom therapy. Examples include a wide variety of techniques and therapeutic formulations, such as tourniquets, snakestones, freezing the bite wound, making incisions to the bite wound and attempting to suck out the venom, applying herbs, or using electric shock therapy in an attempt to neutralize the venom [31,39]. Many snakebite victims seek out traditional healers as their first point of care, and in cases of dry bites, these snakebite victims may get the impression that the traditional treatment successfully cured them of snakebite envenoming. Therefore, treatment methods with no proof of efficacy can gain support, which represents a substantial risk to snakebite victims, who may delay seeking evidence-based medical attention and thereby cause an aggravation of the clinical manifestations of envenoming [31]. 

The reporting protocol for notification of cases of dry bites is not standardized between clinical practices around the world. Among the parameters used in diagnosis, the absence of local or systemic signs and symptoms of envenoming is the main characteristic, which can be accompanied by the presence of fang marks, identification of the venomous snake, absence of laboratorial abnormalities, and an absence of detectable venom levels in body fluids (such as blood and urine). Table 1 reports epidemiological studies on dry bites over the years, and highlight the rates, snake species involved, and diagnostic criteria used to define the bite as dry.

It is likely that dry bite incidence is often underestimated in different studies, due to a number of factors. Among these, there is a trend among some healthcare professionals to not report mild cases and dry bites to the official surveillance system, or to not consider the possibility that a venomous snake can cause a dry bite, in addition to the fact that medical records will naturally be biased towards actual envenomings. The attribution of these cases to injuries by non-venomous snakes can also generate underreporting. Finally, the low demand for medical care due to poor access, ethnic-cultural issues, and preference for alternative treatments, especially in asymptomatic cases, are factors that can also lead to underreporting of dry bites.

Table 1 shows that studies that reported the proportion of dry bites in the literature refer only to bites by viperids and elapids. Since vipers have more advanced venom delivery systems, they are expected to cause fewer dry bites compared to elapids. In our review, it was possible to identify the family of the offending snake for a total of 3025 bites, with a very similar proportion of dry bites for viperids (14.7%) and for elapids (14.5%) (OR = 0.99, CI 95% 0.83–1.18; *P* = 0.886). Thus, contrary to expectations, there was no observed difference in the proportion of dry bites between these two snake families. We suggest that the lack of harmonization in the diagnosis of dry bites among the different studies, as well as potential selection biases that may occur in the search for medical assistance for the different bite cases for different snake families, may explain this result. We highlight that our literature search revealed no information about the participation of other snake families than viperids and elapids in dry bite cases. There are several colubrid genera that possess opisthoglyphous fangs with low-pressure venom delivery systems (such as genera *Boiga*, *Rhabdophis*, and *Chrysopelea*) that often result in dry bites due to the opisthoglyphous fangs. Unfortunately, it seems that research for the clinical characterization of envenomings by colubrids and lamprophiids has been largely neglected, which prevents analysis of their severity grading and proportion of dry bites. 

## 4. Diagnosis of Snake Dry Bites 

The diagnosis of snakebite envenoming can be a complicated task or a comparatively simple one. It all depends on the diversity of venomous snakes in the area, whether the snake was seen, how much time has elapsed between the bite and medical intervention, which clinical manifestations (if any) develop, which tools and assays are available to support diagnosis, and how much experience the treating physician has with clinical management of this highly neglected disease. In general, snakebite diagnosis is often based on a combination of patient history and a syndromic approach, supported by national guidelines and assays assessing e.g., serum biochemistry, coagulopathy (for example by measuring clotting time), and renal function [69]. The diagnosis of a dry bite is generally made retrospectively, once it has been confirmed that the bite was sustained from a venomous snake and that no envenoming occurred. For this reason, the same diagnostic strategies are used for dry bites as for wet bites. 

To confirm the diagnosis of a dry bite, it must be ascertained that the responsible snake was venomous. In cases where the snake was photographed or killed and brought to the clinic, it might be possible to have an expert herpetologist identify it [8,69]. This can rule out bites from non-venomous snakes, even those closely resembling their venomous brethren due to biomimicry. The presence of one or two puncture wounds left by a fang or set of fangs can similarly indicate a bite from a venomous snake, as these marks will often be distinct from those left by non-venomous snakes, which tend to consist of multiple smaller puncture wounds arranged in a semi-circle. While identifying the offending snake as a non-venomous species excludes the possibility of envenoming and dry bite, it should be noted that ‘non-venomous’ does not equate to being harmless. Bite wounds from venomous and non-venomous snakes alike may be contaminated with the snake’s oral flora and can cause non-venom-related illness and discomfort [61]. Snakebites can cause behaviors, which may initially be mistaken for symptoms of toxicity. For example, the bitten person might panic, which can result in hyperventilation, syncope, vomiting, and other clinical features, which can also be observed in some cases of systemic envenoming [8,61,70]. Additional clinical features of dry bites will be discussed in a subsequent section. 

If the herpetologist can positively identify the offending species as a venomous snake, knowledge of the species can help treating personnel better prepare for clinical management of the bite. Knowledge of the species might prepare physicians to source antivenom and give an indication of which type of clinical manifestations can be expected to occur and possibly roughly within which time frame. In the rare cases where the frequency of dry bites is known for the species, the dry bite rate might (at least theoretically) suggest physicians to expect (or not expect) a dry bite. However, even for species with high rates of dry bites, the risk of a wet bite cannot be excluded except retrospectively [69]. 

When it is not possible to visually identify the snake, the diagnosis is often based solely on patient history and a syndromic approach, in which it might be possible to use the clinical manifestations of envenoming, as assessed by laboratorial tests, to infer the type of snake involved. A minimum period of observation is necessary to exclude a snakebite envenoming, in which patients bitten by juvenile species may present no remarkable local manifestations and no signs or symptoms of systemic envenoming, but may present blood incoagulability detected hours after the incident. Thus, laboratorial tests are also important when ruling out envenoming. In most envenomings by viperids, coagulation disorders occur due hemostatically active toxins. The Lee-White clotting time (LWCT) and the 20 min whole-blood clotting test (WBCT20) are simple, inexpensive, and available even in remote health facilities. In general, clotting time tests have had their accuracy estimated in experimental conditions rather than in clinical practice, which may lead to a mistakenly optimistic interpretation of the results in the clinical practice. One study reports that an abnormal clotting time was found in 100% of *Echis carinatus* victims, who presented any sign or symptoms of envenoming, and in the only two cases of dry bites, the clotting time was normal [63,71]. In Manaus, out of 186 patients with *Bothrops* envenoming, 75.3% had prolonged clotting times, and 85.5% had hypofibrinogenemia [72]. Systemic envenoming by juvenile *C. durissus terrificus* resulted in coagulopathy as the main systemic manifestation without other features normally associated with this type of specimens [73]. In Malayan pit viper-bitten patients, an International Normalized Ratio (INR) > 1.155 had a sensitivity of 78.5% and a specificity of 90.3%, while the 20WBCT had a sensitivity of 81.0% and a specificity of 90.3% to assess coagulation abnormality [74]. Thus, such tests constitute valuable tools for the diagnosis of snake envenoming and indication of antivenom therapy.

Tissue injury biomarkers, acute phase proteins, or pro-inflammatory cytokines/chemokines in snakebites may also be used for discriminating between envenomings and dry bites, which could find utility in the clinic. However, few studies exist where the accuracy of these biomarkers has been evaluated. A recent study from Manaus showed that detection of cell-free nucleic acids may be used to discriminate patients envenomed by *Bothrops* and healthy controls [75]. Levels of the inflammatory biomarkers CXCL-9, CXCL-10, IL-6, and IL-10 were higher in *Bothrops*-envenomed patients when compared to the healthy controls on admission [76]. Furthermore, there is a number of interactions of CXCL-8, CXCL-9, CCL-2, IL-6, and IFN-γ and fibrinogen levels in patients bitten by viperids, suggesting these cytokines as biomarkers for this type of envenoming [77]. Detection of other biomarkers, such as lactic dehydrogenase, creatine phosphokinase, and transaminases, mostly available in the hospitals, may possibly be relevant in future well-designed studies.

In addition to the more established laboratory tests and novel biomarkers of envenoming described above, several assays capable of detecting venom components in patient samples have been developed. Many of these are immunoassays, e.g., in the form of enzyme-linked immunosorbent assays (ELISAs) [78,79,80,81,82,83,84,85], lateral flow assays (LFAs) [86,87], impedimetric immunoassays [88], and others [89,90,91,92]. Other researchers have instead explored the potential of utilizing technologies, such as polymerase chain reactions (PCRs) [93,94,95] and enzymatic assays [96], for diagnosis of snakebite envenoming. The differences between these assays in terms of how long time they take to run, user-friendliness, species covered, sensitivity, specificity, limit of detection (LoD), and limit of quantification (LoQ) make them differentially suited for clinical and research use. Despite the availability of several venom detection assays described in the literature, to the best of our knowledge, the only such assay in widespread clinical use is the Snake Venom Detection Kit produced by Seqirus in Australia [97,98,99]. 

Once it has been confirmed that a patient was bitten by a venomous snake, another question to be answered is for how long asymptomatic patients should be observed before they can be discharged. The time from the bite occurs until clinical manifestations of envenoming become noticeable is affected both by the toxin composition and dose of the specific venom, as well as the anatomical site of the bite wound. Some toxins may exert their effects within minutes, while others may require several hours to take effect [8]. Therefore, it is crucial to observe the patient long enough to enable differentiation between delayed onset of toxicity and a dry bite. Usually, clinical manifestations present within 12–24 h (or less) of the bite incident [8]. One study of 360 snakebite patients in Southeast Queensland, Australia, even found that six hours of observation of asymptomatic patients was adequate to exclude elapid envenoming [100], but this claim has since been debated. Another study in Australia proposed a combination of repeated laboratory tests and clinical examination to reliably detect or exclude envenoming at 6 h and 12 h after a snakebite [101]. In South America, an important aspect of *Bothrops* envenomings is that bites by juvenile snakes can cause no local or very mild local manifestations, which can be misdiagnosed as dry bites. However, these cases require a careful examination of the patient’s coagulation parameters, considering the predominance of coagulotoxins in the venom of these immature specimens, which may cause systemic bleeding [102,103]. Figure 2 shows the characteristics of a patient presenting mild, local manifestations and two true dry bites by snakes from the *Bothrops* genus. Once the physician is confident that envenoming can be ruled out and the patient can be discharged, the final diagnosis of a dry bite can be made.

## 5. The Snake Dry Bite Phenomenon 

Different factors can cause snakes to deliver dry bites, some of which are related to humans, and some of which are related to the snakes themselves. 

### 5.1. Snake-Related Factors Involved in Dry Bites

Snake-related factors are responsible for the majority of cases of dry bites (Figure 3), with failure to deliver venom being the main one. Failure to inject venom can in turn be caused by viral infections, physical agents, traumas during defense, excessive pressure on the venom glands (i.e., during manual extraction in captivity), and any other inflammatory response [104]. Therefore, venom gland tissue damage will result in empty venom glands and, consequently, in dry bites. Calcification of fangs and obstruction of the secondary venom ducts can also result in a dry bite. Aged snakes often present these alterations [52]. Mechanical failure resulting in inefficient lunge of the fangs to deliver the venom from the venom sack to the bite site can be also responsible for the phenomenon [69]. 

However, the dry bite phenomenon cannot be reduced to a matter of faulty venom delivery caused by pathologies, which occurs with the high frequency of 20–50% of bites. In fact, most dry bites result from a (deliberate) decision to conserve venom. Snake behavior is age-related, and the age of a snake directly influences the likelihood of it delivering a dry bite. Adults are thought to be far more judicious than juveniles and will therefore more often deliver a dry bite if they perceive that they are under threat, which usually provides them with enough time to escape. In these cases, the dry bite is intentional and could arise in at least one of two ways: (1) The snake could assess the encounter with the target and decide not to activate the extrinsic venom gland musculature, or (2) the snake could activate the venom gland musculature at a level insufficient for venom expulsion [105]. This strategy is called venom metering, which is generally described as a decision on the part of the snake that optimizes energy-related or ecological factors [106]. On the other hand, neonates and juvenile snakes are known to not control venom metering and usually empty their glands during the bites. 

If a victim is lucky, the venom glands of a perpetrating snake might be empty when the snakebite occurs. The likelihood of empty venom glands is affected by the duration since the snake last used its venom and by the age of the snake. Older snakes store more body fat than younger snakes and can replenish venom quicker. Studies show that a delay of around 14 days exists between depletion of venom (due to milking) and maximal rates of venom synthesis [107,108]. 

Another important snake-related cause of dry bites is a misjudgment by the snake pertaining to its distance to the victim. Snakes can interpret this distance using strike-induced chemosensory searching (SICS), which relies on a sustained high rate of tongue flicking [109]. A tiny SICS mistake can mean that the fangs only partially penetrate the prey or that venom is ejected prior to penetration, resulting in a dry bite [110].

### 5.2. Human-Related Factors Involved in Dry Bites

Although less explored, human-related actions are also responsible for dry bites. Snakes instill a deep-rooted fear in many people, which can cause them to make sudden movements in an attempt to escape. Such swift movements at the moment of the bite can result in incomplete penetration of the skin by the snake’s fangs, resulting in a dry bite [110]. Additionally, some clothing materials can obstruct bites. For instance, denim clothing has been demonstrated to reduce the venom release by up to 66% during a bite [111]. Wearing shoes and boots, or any other protective footwear, instead of sandals, can also prevent the venom from being injected into the body of the victim, thereby resulting in a dry bite. 

It has been proposed that, in rare cases, clinical manifestations of venomous wet snakebites can go unnoticed and cause the bite to be misinterpreted as a dry bite due to the natural immunity of some victims [69,112,113]. Some studies have demonstrated the presence of high titers of specific antibodies against toxins from viper venoms in individuals previously and repeatedly exposed to snakebites in the Amazon region, suggesting that there may be some degree of protection against morbidity and mortality in certain individuals [114,115,116,117]. This situation would theoretically imply asymptomatic or oligosymptomatic cases of snakebite envenoming in naturally immunized individuals. However, such cases have never been demonstrated in well-designed cohorts in the region. Likewise, studies have suggested that humans can endocytose venom toxins through antigen presenting cells, such as dendritic cells and macrophages, resulting in the activation of the acquired immunity [113,118]. A study conducted in Nigeria demonstrated that persistent and high levels of IgG antibodies have a protective role against subsequent snakebites [114]. On the other hand, even with detectable titres of antibodies, people who are bitten twice by snakes of the same genus still develop symptoms (e.g., myotoxicity) [119]. Thus, it may be speculated that the potential protective effects of circulating antibodies in recurrent snakebite victims (and self-immunizers) that have developed an adaptive immune response may be dependent on the immunogenicity of the venom toxins of the perpetrating snake species. It is possible that victims bitten by snakes possessing venoms, where the medically most important toxins are low molecular weight toxins, such as three-finger toxins, dendrotoxins, or phospholipases A_2_, are less likely to develop immunity [120,121].

## 6. Clinical Features of Dry Bites: To Treat or Not to Treat?

In the case of dry bites, patients can report pain, usually of light intensity. Local bleeding and erythema can be observed as well, the fang marks may or may not be present, and the systemic signs and changes in laboratory parameters will be absent. It is relevant to note that often, due to the stress caused by the encounter with a snake, the patient can present himself in health services with signs of anxiety, such as tachycardia and tachypnea. It is up to the healthcare professional, at the time of collecting the patient’s clinical history and anamnesis, at admission, and during the observation period of the patient in the hospital environment, to discriminate whether systemic signs could be associated with envenoming (Figure 4). Likewise, the absence of signs and symptoms at the time of admission does not necessarily mean that the patient suffered a dry bite, but that insufficient time has passed for the patient to develop signs of envenoming. Thus, the final diagnosis of a dry bite is always performed retrospectively after confirmation of the absence of local or systemic signs and symptoms even after a follow-up, which may be required up to 12 h later, depending on the aggressing snake species [52,69]. There is a possibility that actions taken by the patient prior to hospital admission, such as the use of anti-inflammatory drugs and even over-the-counter medicines, can delay the appearance of envenoming signs, reinforcing the need for follow-ups. 

In tropical areas, there are many reports of individuals living in places with high exposure to snakebites, who did not seek medical assistance because there was an absence of signs indicating that the patients were at risk. These cases are, however, poorly investigated. For instance, in the Brazilian Amazon, many snakebites are caused by snakes of no medical importance, as their bites only lead to minor trauma with or without fang marks [122]. If the clinical investigation is based only on the report of individuals who have had conditions of this severity during their lifetime, incorrect conclusions can easily be drawn. In some situations, the bitten individual may not have seen the snake or did not observe characteristics that would allow the investigator to identify it. In addition, there are a number of inconsistencies in the nomenclature of some venomous and non-venomous snakes due to similar characteristics (e.g., color and body shape). Therefore, non-venomous snakes are often commonly referred to by the same name as venomous snakes.

Incorrect identification of species happens with specimens of the *Leptodeira* genus and the species *Helicops angulatus*, in the Amazon region, which due to their brownish color are often confused with the venomous species *Bothrops atrox*. In these cases, blood incoagulability is unexpected if the snake is non-venomous, just as in dry bites by pit vipers, but may manifest itself when the bite is from the venomous species. Thus, great care must be taken in the differential diagnosis between dry bites and injuries caused by snakes of limited medical importance. The act of bringing the dead snake that caused the snakebite to the treatment facility, although not recommended, or even photographing it, is important in order to facilitate the identification of a non-venomous snake, and it can be essential in view of the fact that it will prevent the unnecessary use of antivenom, as well as other medicines. However, it does not exclude the need for observation of patients for up to 12 h after the bite.

When a snakebite is confirmed to be a dry bite, antivenom is not indicated. As no venom was injected into the victim, it is expected that there will be only local manifestations of low intensity and limited clinical repercussion, represented mainly by the trauma of the bite itself. The mistaken administration of antivenom in these cases will not bring any clinical benefit to the patient, but may potentially instead lead to early or late adverse reactions resulting from the administration of heterologous immunoglobulins [123]. The bite site must be cleaned with soap and water, and tetanus prophylaxis is an important complementary measure, while antibiotic therapy is only indicated when signs of secondary infection are present. For some types of envenomings, such as those caused by rattlesnakes (*Crotalus* genus) and elapids, local manifestations are generally very discreet, and the systemic signs and symptoms can appear hours after the bite [124,125,126]. Thus, the healthcare professional who assists the case must take care to keep the patient in the healthcare unit under observation, as described above. On the other hand, the clinician may encounter situations in which the benefit of treatment may outweigh the risk of adverse reactions, as in suspected elapid envenomings. As these envenomings can abruptly become more complicated, the practitioner may choose to prescribe the antivenom early [125,126]. With further innovation in envenoming therapy, i.e., the development of recombinant antivenoms based on fully human antibodies [127], the early administration of next-generation antivenoms may become even more warranted, as improvements, in particular to safety, may significantly reduce the risk of adverse reactions [128].

## 7. Conclusions

When biting and injecting their potent venoms into victims, many venomous snakes may impose a serious medical threat to human health. However, as humans are too large to be considered prey for even the largest of venomous snakes, snakes may not always deliver venom when they bite, as they may instead preserve this metabolically costly weapon for predation. Dry bites may cause similar clinical manifestations as bites from non-venomous snakes (or other animals), including inflammation and infection, as well as signs of anxiety, such as tachycardia and tachypnea. As the onset of venom-induced pathophysiology may not occur immediately upon envenoming [129], but instead be delayed until the venom toxins have left the bite site and reached their anatomical site of action, it is of high importance that treating healthcare personnel can provide unambiguous differential diagnosis of dry and wet snakebites. Such diagnosis will determine whether or not antivenom is indicated, as well as it will help inform the treating healthcare personnel about what other supporting measures may need to be arranged in a timely manner to improve clinical outcome. To aid such diagnosis, a more thorough understanding of the both the snake-related and human-related factors involved in dry and wet snakebites is important to obtain, which, in turn, warrants further studies and collection of epidemiological data surrounding both venomous and non-venomous snakebites.

## Figures and Tables

**Figure 1 toxins-12-00668-f001:**
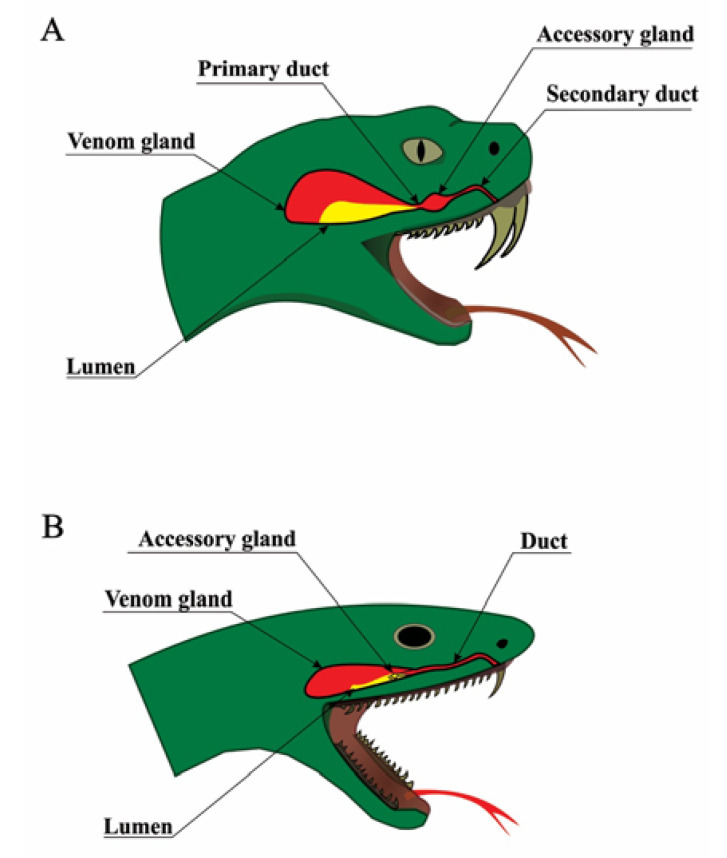
Snake venom delivery systems. Schematic anatomy of snake venom delivery systems. (**A**) Viperidae venom system: The venom gland is triangular and large; the lumen is voluminous and can store high quantities of venom; the lumen forms the primary duct, which is connected to an accessory gland and finally to the secondary duct and the fang. (**B**) Elapidae venom system: The venom gland is oval; the lumen is narrow, and the majority of venom is stored in the secretory cells rather than the lumen; the accessory gland is placed in the distal part of the venom gland and has only one duct.

**Figure 2 toxins-12-00668-f002:**
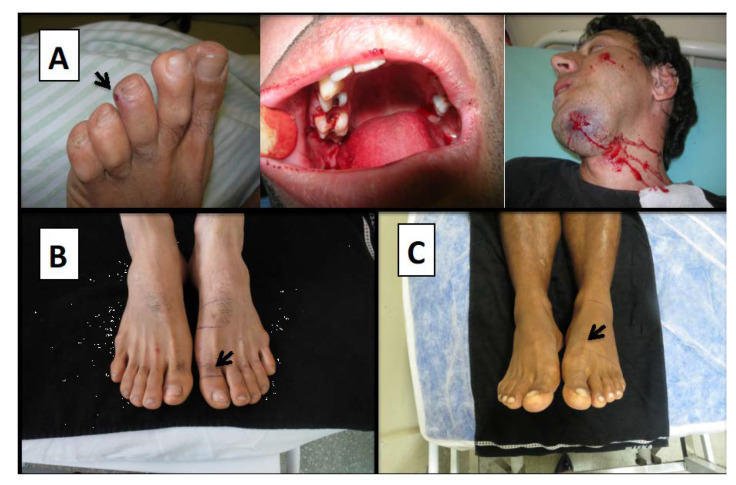
Snakebite cases. (**A**) A case of juvenile *Bothrops jararaca* bite causing mild, local traumatic injury, along with gum bleeding and persistently bleeding chin. Patient also presented hemorrhage in the central nervous system. Patient from the Hospital Vital Brazil, São Paulo—SP. (**B**,**C**) Two cases of juvenile *Bothrops atrox* snakebite presenting only fang marks. No hemostasis disorder was detected. The comparison with the contralateral limb does not show the presence of edema or ecchymosis. The patients had no local or systemic complications at follow-up. Patients from the Dr. Heitor Vieira Dourado Tropical Medicine Foundation, Manaus—AM (ethical approval number 492.892/2013—FMT-HVD Ethical Board).

**Figure 3 toxins-12-00668-f003:**
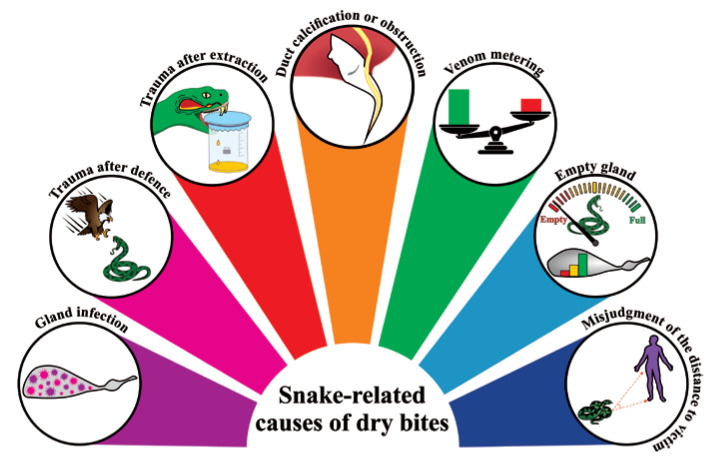
Snake-related causes of dry bites. Schematic representation of the main causes responsible for the dry bite phenomena.

**Figure 4 toxins-12-00668-f004:**
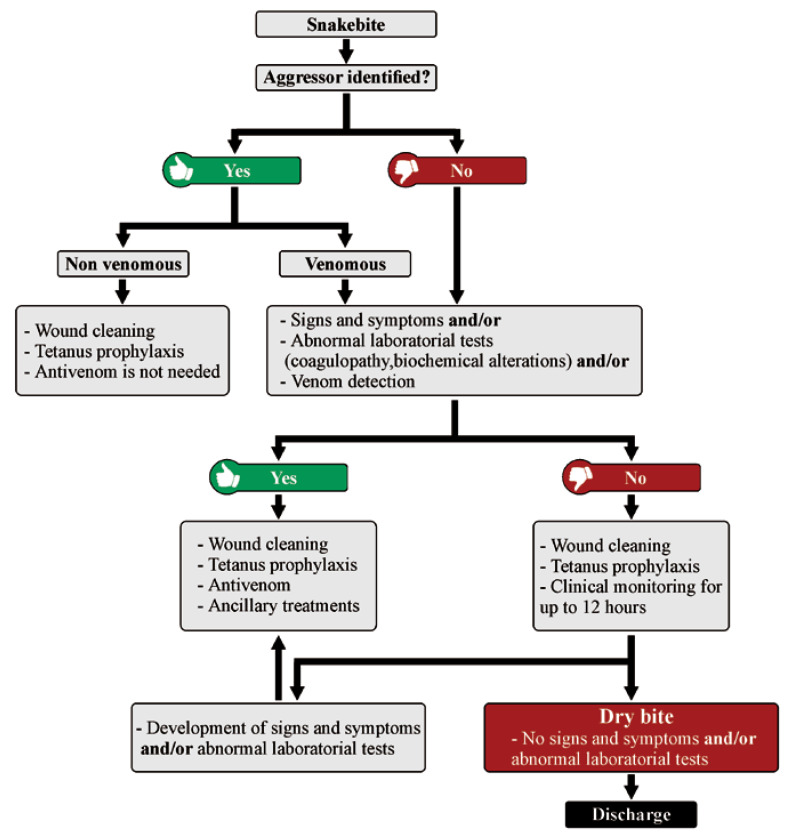
Flow chart on diagnosis and treatment of snake envenomings and dry bites.

**Table 1 toxins-12-00668-t001:** Historical reports of confirmed cases of dry bites: Frequency, snake species, and diagnostic criteria.

Reference	Dry Bite Incidence (%)	Location of Snakebite	Data Collection Period ^#^	Snake Species Involved	Criteria for Dry Bite Diagnosis
Silveira and Nishioka, 1995 [32]	13/40 (32.5%)	Brazil	1992–1994	Lance-headed viper and rattlesnakes	No clinical or laboratory evidence of local or systemic envenoming.
Russell, 1960 [40]	5/22 (22%)	USA	Not reported	Pacific rattlesnakes	No local or systemic signs and symptoms, No lab abnormalities, Presence of fang marks
Campbell, 1963 [41]	29/152 (19%)	Papua New Guinea	1960–1962	*Oxyuranus* sp. *Acanthophis* sp. *Pseudechis papuanus*	No local or systemic signs and symptoms, Presence of fang marks, Snake identified
Reid et al., 1963 [42]	107/212 (50%)	Malaya	1960-1961	*Calloselasma rhodostoma*	Minimal or no local signs and symptoms, Snake identified
Parrish et al., 1966 [33]	335/1,315 (25%)	USA	1958–1959	*Crotalus* sp. *Agkistrodon piscivorus* *Agkistrodon contortrix* Coral snakes	No local or systemic signs and symptoms, Presence of fang marks, Snake identified
Parrish, 1966 [43]	667/2,433 (27%)	USA	1958–1959	*Crotalus* sp. *Agkistrodon* sp. *Sistrurus* sp. *Micrurus* sp.	No local or systemic signs and symptoms, Presence of fang marks
Myint-Lwin & Warrell, 1985 [44]	34/123 (27%)	Myanmar	1983–1985	*Daboia siamensis* *Trimeresurus erythrurus* *Naja kaouthia*	No local or systemic signs and symptoms, Snake identified
Kitchens & Mierop, 1987 [45]	4/20 (20%)	USA	1975–1986	*Micrurus fulvius*	No local or systemic signs and symptoms, Presence of fang marks, Snake identified
Kouyoumdjian and Polizelli, 1989 [46]	1/22 (4%)	Brazil	1986–1987	*Bothrops moojeni*	No local or systemic signs and symptoms, No lab abnormalities (coagulopathy), Snake identified
Curry et al., 1989 [47]	15/146 (10%)	USA	1984–1986	Rattlesnake	No local or systemic signs and symptoms, No lab abnormalities (coagulopathy), Snake identified
Tun-Pe et al., 1991 [48]	91/234 (38%)	Myanmar	1984–1988	*Daboia siamensis*	No local or systemic signs and symptoms, Snake identified
Tibballs, 1992 [49]	10/46 (22%)	Australia	1979–1990	*Pseudonaja textilis* *Notechis scutatus* *Austrelaps superbus* *Pseudechis porphyriacus*	No local or systemic signs and symptoms, Presence of fang marks, No lab abnormalities (coagulopathy), Venom not detected (blood, urine, or washings from the suspected bite site)
Mead and Jelinek, 1996 [50]	32/156 (20%)	Australia	1984–1993	*Pseudonaja* sp. *Notechis* sp. *Pseudechis* sp.	No local or systemic signs and symptoms, Presence of fang marks, No lab abnormalities (coagulopathy), Snake identified
Milani et al., 1997 [51]	1/29 (3%)	Brazil	1975–1995	*Bothrops jararacussu*	No local or systemic signs and symptoms, Snake identified
de Rezende et al., 1998 [52]	5/41 (12%)	Brazil	1994–1996	*C. durissus*	No local or systemic signs and symptoms, Presence of fang marks, No lab abnormalities, Venom not detected (plasma) Snake identified
Tanen et al., 2001 [53]	7/236 (3%)	USA	1994–2000	*Crotalus* sp.	No local or systemic signs and symptoms, No lab abnormalities (coagulopathy + hematological), Snake identified
Kularatne, 2002 [54]	22/210 (10%)	Sri Lanka	1996–1998	*Bungarus caeruleus*	No local or systemic signs and symptoms, Presence of fang marks, Snake identified
Spiller & Bosse, 2003 [55]	31/128 (24%)	USA	2001	*Agkistrodon contortrix Crotalus horridus Agkistrodon piscivorus*	No local or systemic signs and symptoms, Presence of fang marks, No lab abnormalities (coagulopathy)
Bawaskar and Bawaskar, 2004 [56]	1/29 (3%)	India	2001–2003	*Bungarus caeruleus* *Naja naja*	No local or systemic signs and symptoms, Snake identified
Bucaretchi et al., 2006 [57]	1/11 (9%)	Brazil	1984–2004	*Micrurus lemniscatus*	No local or systemic signs and symptoms, Snake identified
Köse, 2007 [58]	4/21 (19%)	Turkey	2004–2005	*Macrovipera lebetinus*	No local or systemic signs and symptoms, Presence of fang marks
Ariaratnam et al., 2008 [59]	4/88 (4%)	Sri Lanka	1993–1997	*Bungarus ceylonicus*	No local or systemic signs and symptoms, Snake identified
Kularatne et al., 2009 [60]	5/20 (20%)	Sri Lanka	1995–1998; 2002–2007	*Naja naja*	No local or systemic signs and symptoms, Presence of fang marks, Snake identified
Walter et al., 2010 [38]	117/838 (13%)	USA	1983–2007	*Micrurus* sp.	No local or systemic signs and symptoms, Snake identified
Warrell, 2010 [61]	5–50%	South East-Asia Countries	Not informed	*Calloselasma* sp. *Daboia russelii* *Echis* sp.	No local or systemic signs and symptoms
Nicoleti et al., 2010 [62]	19/792 (2%)	Brazil	1990–2004	*Bothrops jararaca*	No local or systemic signs and symptoms, No lab abnormalities (coagulopathy), Snake identified
Kularatne et al., 2011 [63]	2/26 (8%)	Sri Lanka	2009–2010	*Echis carinatus*	No local or systemic signs and symptoms, No lab abnormalities (coagulopathy), Snake identified
Kularatne et al., 2011 [15]	1/19 (5%) 2/36 (5%)	Sri Lanka (Central hills)	2006–2008	*Daboia russelii* *Hypnale* species	No local or systemic signs and symptoms, Snake identified
Spano et al., 2013 [64]	5/46 (10%)	USA	2000–2010	Rattlesnake	No local or systemic signs and symptoms, No lab abnormalities (coagulopathy)
Valenta et al., 2014 [65]	51/191 (26%)	Czech Republic	1999–2013	*Vipera berus*	No local or systemic signs and symptoms, Presence of fang marks, Snake identified
Roth et al., 2016 [66]	5/104 (4%)	USA	2009–2011	*Agkistrodon* sp.	No local or systemic signs and symptoms, Presence of fang marks, Snake identified
Silva et al., 2016 [67]	8/33 (24%)	Sri Lanka	2014–not informed	*Bungarus caeruleus*	No local or systemic signs and symptoms,
Bawaskar and Bawaskar, 2019 [68]	1/77 (1.75%)	India	Not reported	*Echis carinatus* *Daboia russelii*	No local or systemic signs and symptoms, Presence of fang marks, No lab abnormalities (coagulopathy), Snake identified

The literature search was performed using the platforms PubMed (pubmed.ncbi.nlm.nih.gov) and Google Scholar (scholar.google.com) using the descriptors “snakebite” added to variations, such as “dry bites”, “asymptomatic”, or “grade 0”. Literature search within references was also performed in order to reach grey literature. ^#^ Represents the approximated period of data collection. In one case, authors only mention “during the past 12 years”.

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
