# Peer review of "Current Knowledge on Snake Dry Bites"

_toxins, 2020, doi:10.3390/toxins12110668_

Round 1

Reviewer 1 Report

The manuscript is interesting and well-written review of snake dry bites.

The manuscript is quite long.

Page 1, Line 27: 100 deaths due to snake bite annually in Europe is not correct information.

Page 5, Line 157: The methodology of literature search for Table 1 should be presented.

Page 14, Line 380 and Page 15, Figure 4: The recommendation for observation of patients after snakebite could not be universal (12-24 hours), since it depends on region and snakes. In Europe, the observation for about a few hours (2 hours) after Viper spp bite should be enough.

Recommendation for 12-24 hours observation at the emergency department could be misunderstood and could lead to unnecessary prolonged management of patients after snakebites.

Overall, this is a very nice review.

Author Response

Ms. Ref. No.: toxins-935830

Title: Current knowledge on snake dry bites

Toxins

Editor-in-Chief: Prof. Dr. Jay Fox

We sincerely thank the reviewers for providing valuable comments. Detailed responses to the reviewer’s corrections/suggestions are described below (in blue).

Reviewer #1:

The manuscript is interesting and well-written review of snake dry bites.

The manuscript is quite long.

Page 1, Line 27: 100 deaths due to snake bite annually in Europe is not correct information.

Response: During the manuscript development, a writing error may have occurred. The correct information was added. Please see page 1, line 27.

Page 5, Line 157: The methodology of literature search for Table 1 should be presented.

Response: The search criteria were added in the legend of Table 1 (page 9, lines 174-178).

Page 14, Line 380 and Page 15, Figure 4: The recommendation for observation of patients after snakebite could not be universal (12-24 hours), since it depends on region and snakes. In Europe, the observation for about a few hours (2 hours) after Viper spp bite should be enough.

Recommendation for 12-24 hours observation at the emergency department could be misunderstood and could lead to unnecessary prolonged management of patients after snakebites.

Overall, this is a very nice review.

Response: This is an important observation once in fact the time for patient observation can vary considering the perpetrating snake, severity of envenoming, and pathophysiology. Our intention was not to generalize the time of observation, but only report that it can go up to 12h. This information is discussed in the manuscript (page 10, lines 261-268). Nevertheless, we modified the sentence in the manuscript in order to clarify that the time for observation can vary depending on the aggressor and that patient observation time can go up to 12h (not standardizing in 12h) (page 13, line 368 and page 14, line 392; and Figure 4).

Reviewer 2 Report

Major comments

The review titled “Current knowledge on snake dry bites” provides an overview of the bites by venomous snakes that result in no envenoming. The topic is important to discuss, and the authors have taken an attempt to give a broad idea about the topic. However, I have some concerns.

If a review to be useful, it should be comprehensive. In several aspects, this article appears ‘superficially written’ and poses the question of “what this article adds more?” in the reviewer's mind, especially given that a recent review on the same topic was published (Naik, "Dry bite" in venomous snakes: A review Toxicon 133 (2017) 63e67)

One issue is that, authors have paid little attention to the taxonomic diversity of venomous snakes and have only focused on viperids and elapids and they have completely neglected venomous colubrid and lamprophiid snakes throughout the manuscript. There are several colubrid genera that possess opisthoglyphous​ fangs with low-pressure venom delivery systems (such as genera Boiga, Rhabdophis, Chrysopelea) that often results in dry bites due to the posteriorly placed fangs. It is surprising that, the authors have not mentioned the term ‘colubridae’ or ‘colubrids’ (largest snake family with several venomous genera!) in the manuscript. The section 2 (Snake venom apparatus and venom production) only describes proteroglyphous and solenoglyphous delivery systems and never mentions opisthoglyphous systems.

Are dry bites common among a particular group of snakes? Since the vipers have more advance venom delivery systems, they are expected to cause less dry bites, compared to elapids and colubrids. The authors have failed to give an insight into this important question.

Although this is ‘general review’ it is always better to use a ‘search criteria’ to find out relevant articles. Since such method was used, the authors have missed some very important articles related to the topic. I strongly recommend the authors to use a search criteria.

A quick pubmed search found following relevant articles missed in the review!!!!

Kularatne et. al. Asian Pac J Trop Med. 2011 Jul;4(7):564-7. doi: 10.1016/S1995-7645(11)60147-8

Kularatne et. al. Trans R Soc Trop Med Hyg. 2009 Sep;103(9):924-30. doi: 10.1016/j.trstmh.2009.04.002. Epub 2009 May 12.

Silva et. al. PLoS Negl Trop Dis. 2016 Feb 1;10(2):e0004368. doi: 10.1371/journal.pntd.0004368.

Thalgaspitiya et. al. Toxicon. 2020 Sep 3;187:105-110.

Chippaux. Bull Soc Pathol Exot. 2002 Aug;95(3):172-4.

Reid. Trop. Med. Hyg. 1975; 78 (5), 106e113.

Specific comments

Terminology: authors have used both ‘envenoming’ (UK English) and ‘envenomation’ (US English) – stick to one.

Page 2, line 56: All venomous snakes

Page 2, line 76: need a reference for the statement

Page 3, line 85: snake venom delivery systems

References: There are many formatting issues (e.g. 36, 37, 50, 63). Please recheck all.

Author Response

Reviewer #2:

The review titled “Current knowledge on snake dry bites” provides an overview of the bites by venomous snakes that result in no envenoming. The topic is important to discuss, and the authors have taken an attempt to give a broad idea about the topic. However, I have some concerns.

If a review to be useful, it should be comprehensive. In several aspects, this article appears ‘superficially written’ and poses the question of “what this article adds more?” in the reviewer's mind, especially given that a recent review on the same topic was published (Naik, "Dry bite" in venomous snakes: A review Toxicon 133 (2017) 63e67)

Response: We very much appreciate the reviewer's comment and understand his/her concern. However, to the best of our knowledge, our review both goes beyond the “state-of-the-literature’ as well as develop new ideas. Although we agree with the referee that there is a previous review exploring the topic, the content explored in ours is much more detailed and updated. For instance, our review presents a detailed table containing epidemiological studies on confirmed cases of dry bites, which was developed through a meticulous search on well-known databases and even using grey literature. I in the diagnostic section, a series of laboratory markers were also discussed in detail as a tool for the diagnosis of mild envenoming versus dry bite. Moreover, this review presents a flow chart on diagnosis and treatment of snake envenomings and dry bites, which could be a very useful tool for clinicians to manage a dry bite. At last, our review is much more updated, with 16 articles from the last three years (2018-2020). This review also discusses real clinical cases assisted by the team, with original illustrations, making the review interesting from a didactic point of view.

One issue is that, authors have paid little attention to the taxonomic diversity of venomous snakes and have only focused on viperids and elapids and they have completely neglected venomous colubrid and lamprophiid snakes throughout the manuscript. There are several colubrid genera that possess opisthoglyphous​ fangs with low-pressure venom delivery systems (such as genera Boiga, Rhabdophis, Chrysopelea) that often results in dry bites due to the posteriorly placed fangs. It is surprising that, the authors have not mentioned the term ‘colubridae’ or ‘colubrids’ (largest snake family with several venomous genera!) in the manuscript. The section 2 (Snake venom apparatus and venom production) only describes proteroglyphous and solenoglyphous delivery systems and never mentions opisthoglyphous systems.

Response: We appreciate the reviewer's comment. We highlight that our literature search has in fact revealed very little information about the participation of colubrids and lamprophiids as dry bite agents. Unfortunately, it seems that research for the clinical characterization of envenomings by these snakes has been largely neglected, which results in difficulties to make good analysis of their severity and proportion of dry bites. We have mentioned subfamilies of the Colubridae genus in the manuscript (Page 14, line 376), as examples of snakes of lesser medical importance in the Amazon. Nevertheless, these subfamilies usually generate mechanical injuries without signs of envenoming. As raised by the reviewer, the anatomical characteristics of these snakes must provide a large number of dry bites, and this discussion was added to the manuscript (See lines 183-189).

Are dry bites common among a particular group of snakes? Since the vipers have more advance venom delivery systems, they are expected to cause less dry bites, compared to elapids and colubrids. The authors have failed to give an insight into this important question.

Response: In our review, it was possible to identify the offending agent for a total of 3,025 bites, with a very similar proportion of dry bites for viperids (14.7%) and for elapids (14.5%) (OR=0.99, IC95% 0.83-1.18; P=0.886). Thus, contrary to expectations, there was no difference in the proportion of dry bites between these two groups of snakes. We suggest that the lack of harmonization in the diagnosis of dry bites among the different studies, as well as selection biases that may occur in the search for medical assistance in the bites by different groups of snakes, may explain this result (See lines 174-182).

Although this is ‘general review’ it is always better to use a ‘search criteria’ to find out relevant articles. Since such method was used, the authors have missed some very important articles related to the topic. I strongly recommend the authors to use a search criteria.

A quick pubmed search found following relevant articles missed in the review!!!!

Kularatne et. al. Asian Pac J Trop Med. 2011 Jul;4(7):564-7. doi: 10.1016/S1995-7645(11)60147-8

Kularatne et. al. Trans R Soc Trop Med Hyg. 2009 Sep;103(9):924-30. doi: 10.1016/j.trstmh.2009.04.002. Epub 2009 May 12.

Silva et. al. PLoS Negl Trop Dis. 2016 Feb 1;10(2):e0004368. doi: 10.1371/journal.pntd.0004368.

Thalgaspitiya et. al. Toxicon. 2020 Sep 3;187:105-110.

Chippaux. Bull Soc Pathol Exot. 2002 Aug;95(3):172-4.

Reid. Trop. Med. Hyg. 1975; 78 (5), 106e113.

Response: We thank the reviewer for pointing out important articles we missed when writing the review. Studies from Kularatne (2009, doi: 10.1016/S1995-7645(11)60147-8 & 2011, doi: 10.1016/j.trstmh.2009.04.002) were included in the Table 1 (historical reports of confirmed cases of dry bites).

- The study from Silva et al. (2016, doi: 10.1371/journal.pntd.0004368) was not included in our review because of an unclear definition of “dry bite”: 8 patients with no neurotoxicity are considered “dry bites” but some of them showed other feature of envenomation (local envenoming for 5 patients for example).

- The full text of the last two references (Chippaux. Bull Soc Pathol Exot. 2002 Aug;95(3):172-4 and Reid. Trop. Med. Hyg. 1975; 78 (5), 106e113) could not be retrieved from any databased platforms and therefore were not included considering the lack of information. Also, the manuscript Thalgaspitiya et. al. Toxicon. 2020 Sep 3;187:105-110 is not a good reference in terms of dry bites. The study brings epidemiological data of the lesser medically important snakebites, most comprising Colubrids (97%), a family with low number of venomous species, in which most snakes involved in the accident are non-venomous.

Specific comments:

Terminology: authors have used both ‘envenoming’ (UK English) and ‘envenomation’ (US English) – stick to one.

Response: We chose the term “envenoming” throughout the manuscript.

Page 2, line 56: All venomous snakes

Response: The term “venomous” was added. Please see page 2, lines 59-60.

Page 2, line 76: need a reference for the statement

Response: This part of the “Snake venom apparatus and venom production” section has been modified.

Page 3, line 85: snake venom delivery systems

Response: The sentence was added two times. Please see page 3, line 91.

References: There are many formatting issues (e.g. 36, 37, 50, 63). Please recheck all.

Response: All references were reviewed.